# Signal Strength and Noise Drive Feature Preference in CNN Image Classifiers

**Max Wolff**[*]
Wesleyan University
mswolff@wesleyan.edu

**Stuart Wolff**[*]
s.wolff1621@gmail.com

## Abstract

Feature preference in Convolutional Neural Network (CNN) image classifiers is integral to their decision making process, and while the topic has been well studied, it is still not understood at a fundamental level. We test a range of task relevant feature attributes (including shape, texture, and color) with varying degrees of signal and noise in highly controlled CNN image classification experiments using synthetic datasets to determine feature preferences. We find that CNNs will prefer features with stronger signal strength and lower noise irrespective of whether the feature is texture, shape, or color. This provides guidance for a predictive model for task relevant feature preferences, demonstrates pathways for bias in machine models that can be avoided with careful controls on experimental setup, and suggests that comparisons between how humans and machines prefer task relevant features in vision classification tasks should be revisited.

## 1 Introduction

Deep neural networks (DNNs) can be used for a wide range of tasks, yet we do not yet have a fundamental understanding of how DNNs actually perform many of these tasks. In this paper we focus on image classification and seek to explain why image classifiers select certain features of the input space over others.

Feature preference in CNNs has been explored through prior research and the results often suggest that machines classify images very differently from humans. Adversarial example research has shown that CNN classifiers can be easily fooled by small and imperceptible (at least to humans) manipulations to inputs. Ilyas et al. (2019) suggest that CNN classifiers key off of widespread non-robust brittle features that are present in the dataset but imperceptible to humans leading to a misalignment with human expectation. Jacobsen et al. (2019) suggest that one type of adversarial vulnerability is a result of narrow learning and is caused by an overreliance on a few highly predictive features in their decisions rendering the models excessively invariant. They suggest this is a result of cross-entropy: maximizing the bound of mutual information between the labels and the features. However, they do not offer an explanation for why models lock in on some highly predictive features and ignore others. Hendrycks & Dietterich (2019) developed a benchmark to test the robustness of CNNs to perturbations and corruptions. Rusak et al. (2020) pointed out that the human visual system is generally robust to a wide range of image noise but that machine models strongly degrade with various types of unseen corruptions.

Bias in machine models is a very important topic that is being actively investigated. Geirhos et al. (2019) designed a set of cue conflict experiments to compare how machines and humans classify ImageNet objects. When ImageNet-trained CNNs were fed images with conflicting shape and texture

---

[*]Equal contribution.

3rd Workshop on Shared Visual Representations in Human and Machine Intelligence (SVRHM 2021) of the Neural Information Processing Systems (NeurIPS) conference, Virtual.

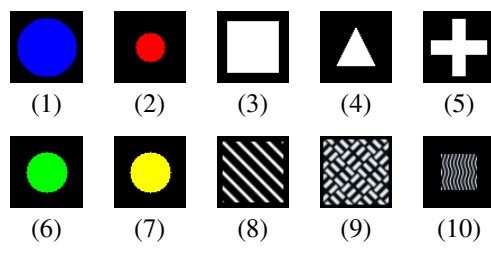

Figure 1: Example features (contained in a 64x64 box) used in the pairs matrix experiments. (1) blue circle (2) red circle (3) square (4) triangle (5) plus (6) green circle (7) yellow circle (8) banded (9) blocky (10) wavy

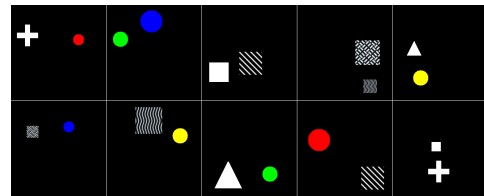

Figure 2: **Top row:** cue conflict examples from Pairs Matrix 1. **Bottom row:** cue conflict examples from Pairs Matrix 2.

features, the results showed that while humans preferred to classify these cue conflicts according to shape, the machines preferred to classify the images according to texture, which was described as texture bias. Subsequent research by Hermann et al. (2019) showed that by augmenting the training data of an ImageNet CNN, they were able to increase shape bias, and concluded that texture bias is not an inductive bias of the underlying model. Another machine bias was identified by Shah et al. (2020)–simplicity bias–which they described as the tendency for neural network classifiers trained with gradient descent to form the "simple" decision boundary before a more robust "complex" one. They suggest that simplicity bias can limit the robustness and generalization of machine models. Geirhos et al. (2020) outlines various types of DNN biases and unintended failure examples which they describe as shortcut learning. This occurs when solutions are found to tasks that are not the result of learning the intended human-like solution. Hermann & Lampinen (2020) used synthetic image datasets and found that the "easiness" and "predictivity" (how often the feature predicts a class) were positively correlated with a CNN's preference for that feature.

Most of the previous work on feature preference in CNNs has shown that machine models will prefer "easy" or "simple" features over more "complex" or "difficult" features, and this can lead to errors, biases, and misalignment between machine and human vision. In this work, we present a foundation for what actually makes task relevant features "harder" or "easier" for CNNs to identify (and ultimately use) in classification tasks.

**Contributions**

- CNNs will prefer task relevant features that are represented with larger signal—larger number of pixels—over task relevant features that are represented by smaller signal—smaller number of pixels.

- CNNs will prefer task relevant features that are represented with lower noise. We identify several feature attributes that increase noise and therefore lower preference including: deviation, overlap, and predictivity.

- CNNs show no strong preference between color, shape, or texture—feature equivalency—when signal and noise are carefully controlled.

## 2 Methods

To understand what features will be preferred by a CNN image classifier in a controlled environment, we start with ten basic, synthetic features, which can be seen in Figure 1. There are 3 different "shape" features, 3 different "texture" features, and 4 different "color" features. From these ten features, we create 45 different classes, each representing a different combination of two of the ten features in the same image, against a black background. The 45 combinations are then separated into nine different "pairs matrix datasets" with five classes each, where each feature appears in each dataset exactly once. We then train a ResNet-18 He et al. (2015) on each set using a modified version of the torchvision ImageNet training script.

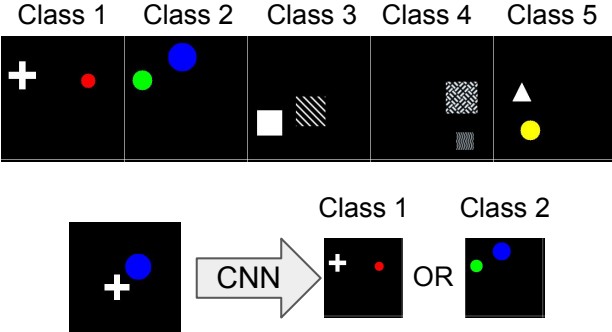

Figure 3: We designed a pairs matrix cue conflict classification experiment to test feature preference. In one of the pairs matrix experiments, a CNN was trained to classify the five classes of feature combinations in the top row. If the CNN chose "Class 1" rather than "Class 2" in the image shown in the bottom row, then the "plus" feature is preferred to the "blue circle" feature. As seen in Section 3, sufficiently increasing the size of the "blue circle" feature can shift the CNN's preference towards the "blue circle" feature from the "plus" feature.

After training, the classifiers were tested on cue conflict images. Specifically, we measure the trained classifiers' responses to examples of all of the 45 combinations of features, regardless of whether the classifier was trained on those combinations (see Figure 3). Then, we recorded the number of times a feature's class was predicted by the classifier, and divided it by the total number of times a feature appeared in the cue conflict test set. The results were then aggregated across all classifiers and datasets to generate a feature preference ranking. The more a feature's class was predicted in cue conflict images by a classifier, the more that feature was generally preferred. By manipulating the qualities of the original ten features, we were able to quantitatively measure the effect that these manipulations had on how much a feature was preferred by a classifier.

We render 300 images per class for training, 100 images per class for validation, 100 images per class for testing, and 100 images per feature combination during cue conflict testing. We create one dataset per set, and average preference results across five training runs. All models are trained for 90 epochs with SGD using learning rate 0.1, which gets decayed by 0.1 at epochs 30 and 60, and with weight decay 0.0001. Images are normalized by ImageNet per-channel means and standard deviations. Features in training images are placed within a 192x192 box, padded with 32 pixels on each side, and the 256x256 result is randomly cropped into a 224x224 image. This procedure is used for experiments detailed in Section 3.

## 3   Factors That Influence Feature Preference

In this section we present the results from our pairs matrix experiments and explore the factors that either increase or decrease feature preference.

**Pixel Count**

We find that when variables are controlled, there is a high correlation between the number of pixels used to represent a feature and that feature's preference. Specifically, we construct a pairs matrix that contains three elementary shapes (triangle, square, and plus), four different colors contained within a circle (red, green, blue, and yellow), and three different textures (blocky, wavy, and banded). We vary the pixel count for each of the ten features and test for preference.

Within these ten features, we create three feature groups, where each group contains features with approximately the same number of pixels: one group contains one shape, color, and texture with a large number of pixels; one group contains one shape, color and texture with a small number of pixels; and one group contains one shape, color, and texture with a medium number of pixels. We have one spare color that is inserted into the medium pixel group.

We observe a strong correlation between the number of pixels that define a feature and the average preference given to that feature, which can be seen in Figure 4. Large and small feature groups are

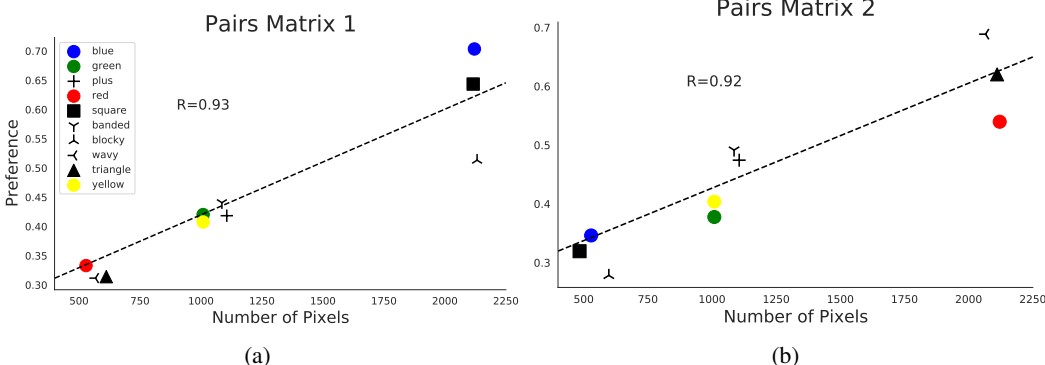

(a)    (b)

Figure 4: Feature preference is linearly correlated with pixel count. Feature preference is averaged across five runs of a pairs matrix experiment. The dataset included four colors, three shapes, and three textures. In (a), the color, shape, and texture with the largest number of pixels showed the highest preference and the features with the smallest number of pixels were least preferred. In (b), the features from (a) that had the largest number of pixels were reversed with the features that had the smallest number of pixels (the middle group was left unchanged) resulting in a reversal in feature preference. This shows that the ResNet-18 does not prefer any feature type (color, shape, or texture), implying feature equivalency.

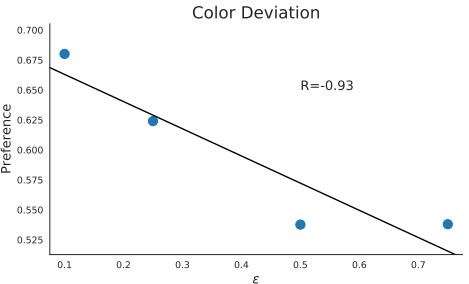

Figure 5: Feature preference is linearly correlated with color deviation. The hue of the blue circle feature dataset in the pairs matrix experiment is modified with $U(-\epsilon, \epsilon)$ during both training and testing. The pixel count was held constant during this experiment. Results show that increasing deviation of a feature will decrease the preference.

Figure 6: Feature preference is linearly correlated with overlap between two features. In this experiment, the blue circle feature of our pairs matrix is linearly interpolated towards the green circle feature. Higher values of blue-green interpolation indicate higher values of color hue overlap between the two features. Pixel counts of all features were held constant during this experiment. Results show that increasing overlap between two features will decrease the preference for both features.

reversed in Figure 4 (a) and (b), but the correlation between the number of pixels and preference remains. Moreover, this correlation holds across various feature types including shapes, colors, and textures. This shows that for CNN classification the number of pixels that represent a task relevant feature defines signal strength, which in turn drives feature preference. Importantly, when signal strength is normalized, the CNN shows feature equivalency; no preference for color, shape, or texture features was observed.

**Deviation**

Deviation on task relevant features is a common characteristic of popular classification datasets such as MNIST and ImageNet, and increasing deviation will make a classification task more difficult. For example, a handwritten seven might exist in two variations: with a horizontal line through the

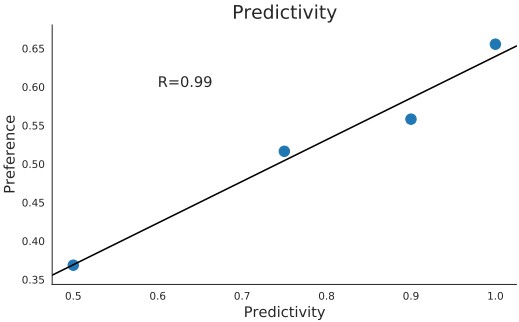

Figure 7: Feature preference is linearly correlated with predictivity (presence). The predictivity of the blue circle feature of the pairs matrix is varied. Pixel counts were held constant during this experiment. Results show that decreasing predictivity (presence) will decrease the preference for a feature.

center and without; a handwritten digit classification model would have to learn both. Thus, it follows that adding deviation to a task relevant feature will likely make the feature less preferred by a CNN, as it increases the difficulty for a model to capture that feature. Once we established controls for signal strength above, we moved to the second set of experiments which focused on quantifying the effects of different types of deviation on task relevant feature preference. For these experiments, we conducted pair matrix experiments similar to what was used for signal strength, but we added deviation to the hue of a color circle during training. As displayed in Figure 5, the amount of hue deviation added to a color during training is linearly correlated with the preference of that feature.

### Overlap

We also hypothesize that inter-class feature overlap has a negative effect on feature preference. In this experiment, we linearly interpolate the blue color circle to the green color circle, and keep their pixel counts the same. All aspects of every feature in Pairs Matrix 1 are kept constant, except for the blue circle feature which we bring down to the medium pixel feature group. Like deviation, interpolating two features together linearly decreases the preference of the CNN for both features, as can be seen in Figure 6. If class relevant features are interpolated together, then each feature loses the predictive power they hold for their respective classes.

### Predictivity

Drawing from experiments described by Hermann & Lampinen (2020), we conducted experiments where we varied the frequency that a feature is present in its given class for each set in a pairs matrix experiment. For example, if the predictivity of a feature is set to 50%, the feature is present in only 50% of training instances for that class. As seen in Figure 7, decreasing a feature's predictivity will cause a decrease in the preference in a pairs matrix experiment. Like inter-class overlap between features, decreasing a feature's predictivity decreases that feature's predictive power for its respective class, which will result in a lower feature preference relative to other predictive feature options.

## 4  Discussion and Conclusion

The results of these experiments show that CNNs exhibit signal preference. For vision recognition, the base signal of a feature may be described as the number of pixels used to represent that feature. Increasing the signal will increase the preference for that feature over other task relevant features with lower signal assuming all other variables are controlled as shown in Figure 4. The results also show that increasing noise factors which make the feature more difficult to capture or decrease the predictive power of the feature will lower feature preference as shown in Figures 5, 6, and 7. For vision recognition, noise includes 1) deviation between feature instances within a class 2) inter-class overlap between a task relevant feature and another task relevant feature, and 3) predictivity (presence or absence of the feature on some class instances in the dataset).

While performance of CNN image classifiers has surpassed the capability of humans, we still lack a fundamental, formal understanding of how CNNs perform vision tasks. Gaining this understanding

will not only inform the development of safe and interpretable vision models, but also has the potential to provide insight into how human vision functions as well.

Essential to understanding visual processing systems is a predictive model for such a system's preference for features in its input space. Based on the results of this paper, we propose a model for feature preference generally expressed as:

F(pr) = Signal (pixel count) - Noise (Deviation + Overlap + Predictivity)

Having a predictive model for CNN feature preference has the potential to help in a range of research topics including interpretability, new data augmentation strategies and training objectives to expand the range of task relevant features that are included thereby potentially improving generalizability, robustness, and accuracy for some tasks and datasets. While working towards the predictive model we also found a pathway to ascribe biases to machine models that might be an artifact of some experimental setups. In particular, if Signal and Noise are not carefully controlled for, it is possible to find or mask feature preferences based on the test set that is used without needing to make changes to a trained model or dataset. For example, in our synthetic dataset, we can easily show preference for colors, shapes, or texture features by simply making the desired feature be defined by more pixels in the test set. We can also shift the feature preference by adding or removing deviation, overlap, or predictivity.

We also consider the impact that these experiments might have on the comparisons that have been made between machine and human vision. When tasks and datasets are carefully controlled for Signal and Noise, we expect that feature preferences of machines and humans move closer in alignment, but this must be tested experimentally, and should be explored in future work. Future work should also test for the extensibility of these results across other tasks (including unsupervised objectives), datasets, data augmentation strategies, and architectures. The results may also inform DNN learning theory.

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

# A   Revisiting Texture Bias

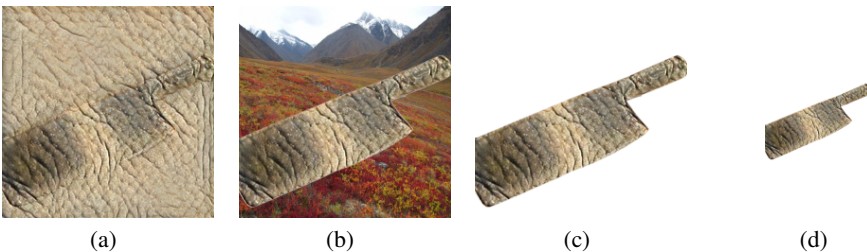

|       | (a)       |       | (b)       |       | (c)       |       | (d)       |

Figure 8: Modifications applied to cue conflict stimuli, from lowest to highest shape preference. (a) no modification (b) exterior mask then landscape (c) exterior mask (d) exterior mask then resize (50%).

| MODIFICATION | SHAPE BIAS | TEXTURE BIAS | ACC. |
|---|---|---|---|
|  | 21.4 | 78.6 | 65.8 |
| FULL SHAPES | 23.2 | 76.8 | 65.7 |
| LANDSCAPE | 55.2 | 44.8 | 58.8 |
| MASK | 66.3 | 33.7 | 66.3 |
| FULL SHAPES (M) | 72.8 | 27.2 | 66.5 |
| RESIZE (50%) | 87.7 | 12.3 | 57.1 |
| RESIZE (25%) | 89.1 | 10.9 | 42.7 |

Table 1: Effect of each modification to the cue conflict stimuli of Geirhos et al. (2019) in order of increasing shape bias. Acc. refers to the percentage of stimuli that were classified according to either shape type or texture type. Full shapes (M) refers to full shape features that had an exterior mask applied to them.

## A.1   Methods

To measure the texture bias of ImageNet-trained CNNs, we follow the procedure outlined by Geirhos et al. (2019). We use the style transfer shape-texture cue conflict and silhouette stimuli open-sourced by the authors. We keep the texture bias measurement procedure exactly the same, and modify only the test images. Details and results of these experiments are contained in Section A.2. All texture bias measurements were recorded using a `torchvision` ResNet-50.

## A.2   Results

Geirhos et al. (2019) showed that ImageNet-trained CNNs will classify an object according to its texture rather than shape in what they described as texture bias. The result is intriguing and important since it suggests that ImageNet trained models seem to classify objects quite differently from humans. When we reviewed the cue conflict experiment of Geirhos et al. (2019) the test images appeared to contain a disproportionate amount of texture signal compared to shape signal which could potentially skew the results towards texture bias. The reason for this is 1) during the style-transfer process, a texture will get mapped over the entire image, while the shape remains fixed in a portion of the image

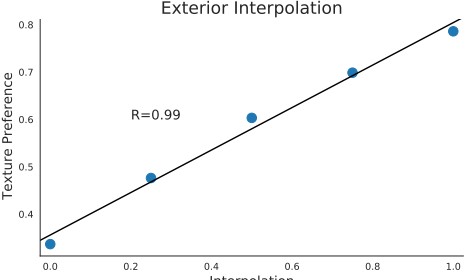

Figure 9: ImageNet-trained ResNet-50 feature cue conflict texture preference is linearly correlated with the signal strength of the cue-conflict background. In this experiment, the background of each cue conflict image was masked to white, and then interpolated towards the original cue-conflict background. Higher values of interpolation indicate higher similarity to the original cue-conflict background. As the background becomes more similar to the texture cue it increases the overall signal of the texture feature and results in increased texture bias.

Figure 10: ImageNet-trained ResNet-50 texture preference is linearly correlated with masked cue conflict feature size. As the feature size increases the texture signal (surface area at $R^2$) to shape signal ($R$) ratio increases which results in increasing texture bias.

resulting in a large pixel count imbalance between texture and shape (in favor of texture) and 2) the stye-transfer process often distorts shape information. We revisit the experiment to see if our hypothesis on signal and deviation can be used to gain increased control over the cue conflict test set and then study whether the conclusion on texture bias still holds. In the experiments presented in this section, we modify the texture-shape cue conflict images used in Geirhos et al. (2019) so that the texture and shape signals (number of pixels) in each feature are varied in a controlled manner relative to each other and the cue conflict preference is measured.

**Masking**

To mitigate the effect of the texture-shape signal imbalance in the cue conflict experiment, we mask the background around the shape in each image. This eliminates texture signal outside of the object, and increases shape signal by increasing the contrast between the background and silhouette of the shape.

We find that after performing this masking operation, ImageNet-trained CNNs show a preference for shape (66% shape bias, 34% texture bias) following the testing process that Geirhos et al. (2019) described. While Geirhos et al. (2019) and Baker et al. (2018) conducted a similar experiment that still resulted in a texture bias, there was a key difference that we believe led to a different result from our experiment. In experiments performed in Geirhos et al. (2019) and Baker et al. (2018), a texture was mapped onto the silhouette of a shape whereas in our experiment the texture was mapped onto the object using style-transfer. Silhouettes do not contain all the shape signal, as there is some shape information contained within an object that gets removed in a silhouette representation. In contrast, the style-transfer process preserves these important shape signals, so we believe that our experiments created a more accurate comparison between shape and texture preference in ImageNet-trained CNNs.

**Resizing**

Since a texture gets mapped over the entire area of an object while shape information is contained in edges, we hypothesized that decreasing the size of the masked version of the texture-shape cue conflict test image would further increase shape signal relative to texture signal since the texture signal will decrease proportional to dimension squared whereas shape will drop linearly as object size decreases. Indeed, there is a strong negative correlation between object size and CNN texture preference (Figure 10).

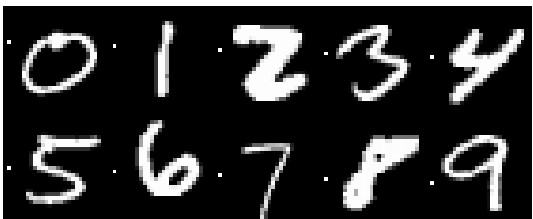

Figure 11: Examples from the Binary shiftMNIST dataset.

**Using Only Full Shapes**

In the cue conflict experiments of Geirhos et al. (2019), some of the shape features in the test set images have incomplete shapes (e.g. the tip of a knife may be missing) whereas texture features are generally complete in all test images. This should further reduce the amount of shape information contained in a test image, and thus result in lower shape preference. To test this effect, we removed (manually selected) cue conflict test images with incomplete shapes from the cue conflict test set, and re-measured texture-shape preference. Interestingly, we saw no significant change in texture preference using the original style-transfer images with incomplete shapes removed, but we did observe an increase (8.5%) in shape preference after exterior masking. This further illustrates the impact of background texture signal in the original style-transfer cue conflict test images.

**Discussion**

Informed by the results outlined in Section 3, we demonstrated increased control over test images in the texture-shape cue conflict experiments proposed by Geirhos et al. (2019). In Hermann et al. (2019), the authors were able to shift the CNN from texture bias to shape bias by augmenting the training data of the ImageNet model but we were able to shift from texture bias to shape bias strictly by changing the test images while using exactly the same ImageNet-trained model. We believe follow-up experiments with full control over signal, deviation, overlap, and predictivity in the test images across all classes are needed to accurately quantify the level of texture bias in ImageNet-trained CNNs.

## B    Revisiting Excessive Invariance

### B.1    Methods

The procedure for this experiment largely follows experiments done in Jacobsen et al. (2019). We construct three different test sets, and one training set. For test sets, one is the unmodified MNIST test set, one contains MNIST digits with location-based class-conditional pixels, and the last only location-based class-conditional pixels. The training set is constructed by placing the location-based class-conditional pixel next to MNIST digits.

To extract segments of the training set with a given amount of deviation, we trained a ResNet-20 classifier on the original MNIST dataset, and created a new training set containing images closest, within a given percentage, to the per-class mean in latent space.

We trained models using a ResNet-20 optimized using SGD with learning rate 0.1 (decayed by 0.1 at epochs 30 and 40) for 50 epochs, and with weight decay 0.0001. A random 55,000 images from the total 60,000 MNIST pool of training images was used for training and the rest was left for validation. Our model achieved 99.55% clean accuracy on unmodified MNIST. After training, a model is tested on all three test sets. We average all results across five training runs.

### B.2    Results

Jacobsen et al. (2019) demonstrated in their Binary shiftMNIST experiment that by adding a single location-based, class-conditional pixel to an MNIST dataset during training but removed at test time, classification accuracy dropped from 100% to 13%–just slightly above random guessing. Clearly the

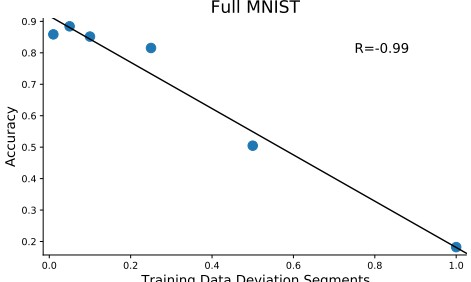

Figure 12: In the Binary shiftMNIST experiment 1, the classification accuracy for a test set with just the MNIST digits increases as the deviation of the MNIST digits in the Binary shiftMNIST training set decreases. The training set includes a location-based, class-conditional pixel for each class and the MNIST digits segmented by deviation.

Figure 13: In the Binary shiftMNIST experiment 2, classification accuracy for a test set with just the location-based pixel decreases as the deviation of the MNIST digits in the Binary shiftMNIST training set decreases. The training set includes a location-based, class-conditional pixel for each class and the MNIST digits segmented by deviation.

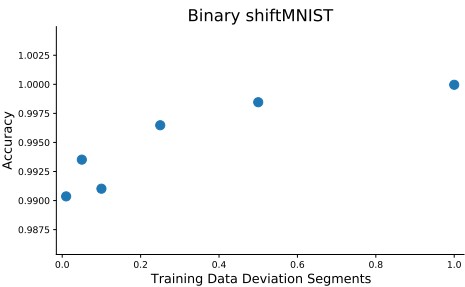

Figure 14: In the Binary shiftMNIST experiment 3, classification accuracy for a test set with both the location-based pixel and the MNIST digits remains relatively constant as the deviation of the MNIST digits in the Binary shiftMNIST training set decreases. The training set includes a location-based, class-conditional pixel for each class and the MNIST training digits segmented by deviation.

model had developed a strong preference for the pixel feature over the MNIST digit feature when both were equally predictive.

Given our results above on feature preference, we wanted to see if we could increase the preference for the MNIST features in the Binary shiftMNIST experiment by decreasing the deviation of the MNIST features. We first separated the MNIST dataset into class segments based on the deviation from the mean in latent space of a ResNet-20 MNIST classifier. The deviation segments ranged from 1% (all class instances were within 1% of the mean in latent space and then replicated), 5%, 10%, 25%, 50%, and 100% (the full MNIST dataset).

We then trained a ResNet-20 on each of these modified MNIST datasets, augmented with a single location-based pixel for each class like the Binary shiftMNIST experiment. Results from this experiment, Binary shiftMNIST experiment 1, can be seen in Figure 12. We then tested this same model by removing the MNIST digits but leaving the location-based pixels in Binary shiftMNIST experiment 2 (results shown in Figure 13). In Binary shiftMNIST experiment 3 we tested the same model with both the location-based pixels and MNIST digits present in the test images (results shown in Figure 14).

## B.3 Discussion

When the full MNIST dataset was trained with the location-based pixel features in Binary shiftMNIST experiment 1, we were able to reproduce the result from Jacobsen et al. (2019) where the model was unable to accurately classify the MNIST features when the location-based pixel features were

removed from the test set. However, as we decreased the deviation in the MNIST training digits, we were able to progressively increase the classification accuracy for the MNIST features suggesting that the model began including the MNIST features in its feature representations. When the MNIST training features were within 5% of mean, the model was able to classify the MNIST test set with a relatively high accuracy. In Binary shiftMNIST experiment 2, we found that the classification accuracy decreased as the deviation in the MNIST training digits decreased showing that the model became invariant to the location-based pixels once the deviation in the MNIST digits was sufficiently low.

When the model was tested with both location-based pixels and MNIST digital we found that it was able to classify at a high accuracy for all the deviation segments. We believe that this shows that the model was learning 1) just the location-based pixels when the MNIST digits had high deviation, 2) just the MNIST digits when they had low deviation because of the much larger signal provided by the MNIST features and 3) a subset of both the location-based pixels and MNIST digits in an entangled representation for the mid level deviation segments (i.e. not able to accurately classify either feature separately). This shows that the model we developed in Section 3 holds predictive potential for other results in machine learning.

