# OpenReview forum: "Signal Strength and Noise Drive Feature Preference in CNN Image Classifiers"
_NeurIPS.cc/2021/Workshop/SVRHM — SVRHM 2021 Poster_

### Official Review · Reviewer_q5Um · 2021-10-14
**An important set of observations on what drives feature preference in CNN image classifiers**

**Rating:** 8
**Confidence:** 3

**Review:**

I enjoyed reading this paper, and think it offers a number of important observations for understanding feature preference in CNNs. It also very nicely fits into the scope of this workshop, as understanding differences/similarities between human and machine image classification is a clear motivator. Additionally, the attention to experimental design shows that backgrounds in experimental science can lead to valuable contributions to the study of deep neural networks. For these reasons, I recommend the paper to be accepted as is, but I believe addressing points 1-3 below will make it a stronger, more impactful paper.

Strengths:
1) Problem of interest was well motivated;
2) How the results fit into existing literature was well explained;
3) Interesting results that were motivated and explained well;
4) The importance of considering experimental design is nicely illustrated;
5) What the future directions this work encourages/inspires, and why those will be interesting, was well explained.

Weaknesses:
1) "When signal strength is normalized, the CNN shows feature equivalence" [line 104] was not, to the best of my knowledge, explicitly shown in the paper. I think this is an extra interesting point, and while I believe it, it would be nice to see some experimental confirmation of this;
2) Figures 1, 2, 7, 8, and 10 were not referenced in the text, nor were the appendices A and B. I think including references to them throughout the text would make the reading a little more clear;
3) A minor point, but I didn't understand why the feature groups were reversed in Fig. 3 (a) and (b). Is there some underlying reason to this, or was it an experimental design choice?

Lastly, I believe there is a small typo on line 94 (should be "the [same] number of pixels".

---

### Official Review · Reviewer_QTRY · 2021-10-19
**Interesting project that needs some clarification regarding the design of experiment and stimuli**

**Rating:** 6
**Confidence:** 3

**Review:**

The authors explore the features that DNNs rely on most heavily when classifying objects, and the factors that influence which features are most important.  The research question is interesting and the method is appropriate (I think).  I add “I think” because I cannot follow the design of the stimulus set starting on line 62 – I find the entire paragraph hard to follow.  More detail needs to be provided, including explaining more explicitly what “pairs matrix datasets” are.  Apart from expanding the text, a detailed figure would help, and a more detailed caption for Figure 1 is required as well.  Overall, the paper is well written, but when it comes to this critical part, I just could not follow.  That said, the logic seems good, so I think it is just a matter of clarifying this.

The authors note that the size of the feature is critical when other factors are controlled.  That makes sense, but it is perhaps worth citing recent work that highlights how a single pixel embedded in CIFAR images that is diagnostic of a category can be used by CNNs (while ignoring the full image).  See: Malhotra, G., Evans, B. D., & Bowers, J. S. (2020). Hiding a plane with a pixel: examining shape-bias in CNNs and the benefit of building in biological constraints. Vision Research, 174, 57-68.  This finding shows very small features can be more important than larger features when the small features are more reliable.
Are the Appendices intended to be part of this paper?  They seem almost unrelated and much more detail is needed if they are to be usefully included.

---

### Official Review · Reviewer_Z45s · 2021-10-26
**Interesting research question, but the research and paper structure is unclear**

**Rating:** 4
**Confidence:** 4

**Review:**

The study presented in this paper aims to better understand which features of images play a more significant role in determining the CNN final classification decision. The research stems from a line of research from Geirhos and colleagues who showed that CNNs tend to have a bias towards the texture whereas humans tend to have a bias in favour of objects silhouette when having to decide between contrasting clues. Geirhos and colleagues test this bias by typically training a CNN on an image dataset and then presenting chimera test images composed by the texture of an object and the silhouette of another object. Then they measured whether the image is classified based on the silhouette or the texture.
In the present paper, the authors aim to test the relative importance of a series of characteristics in a synthetic dataset: colour, shape, and texture.
The authors report that when controlling for signal to noise ratio, none of the above features was preferred over the others when the CNN had to decide using contrasting clues.
The idea of better understanding the inner works of a CNN and what feature of an image are the CNNs representing is a significant and impactful goal. And finding that no feature between colour, texture and shape is preferred when signal to noise ratio is of interest.
However, I have some queries about the methods and the paper structure:
* The authors state that their aim is to test colour, shape, and texture and, in the methods, they explain how they constructed the synthetic dataset. However, they never present the results for these features. In the results section the results from other tests and experiments are presented.
* A ResNet-18 is trained on what, from the description in the methods, seems to be their standard dataset composed of synthetic images of different shape, colour, and texture. The network is then tested on a conflict task in which two conflicting images are presented together. However, in the experiments presented in the results different features have been manipulated. Specifically, the authors present experiments in which they manipulated pixel number, deviation, overlap, and predictivity of each feature. The results reported show that when a feature encompassed more pixels, when the signal to noise ratio was higher, and when a feature was more uniquely related to the object (less deviation, less overlap, and higher predictivity) that feature was preferred when the CNN was forced to choose between contrasting cues. How were these properties expressed on the train dataset? Was the network re-trained for each experiment?
* The authors in the discussion propose a model to account for these features presented in the results section but I did not understand how the model was built and whether it was tested. And if it was, how was it tested and what were the results?
* In the discussion the authors claim that they would expect feature preference between machines and humans to move closer in alignment if the signal to noise ratio was carefully controlled. I am not sure I understand this argument, especially considering that humans are able to successfully solve classification tasks even in bad visual conditions.
* In the appendix two more experiments are proposed and they seem somewhat unrelated to the ones proposed in the main paper. In the first, the authors present an alternative to the experiment of Geirhos and colleagues. Specifically, they propose to use a different style transfer function, but it’s not clear what they wanted to maximize. First, it seems that the new style function makes it so that the number of pixels that form the shape are similar to the number of pixels that form the texture. It is claimed that the new style transfer function preserves shape information contained within the object which gets lost when focusing on the silhouette.
* In the last experiment presented in the appendix a ResNet-20 was trained on the MINST and some modifications with the aim of investigating the effect of location-based pixels and deviation.
* It is not clear why for each part of the paper (main paper, appendix experiment 1, appendix experiment 2) a different network was used (ResNet-18, ResNet-50, and ResNet-20 respectively).

Overall, though I praise the clear effort that the authors put in this work, I feel the presentation of the methods and the results could be clearer.
My advice would be to focus on one question (either colour/texture/shape or predictivity/overlap/deviation/pixel count) and write a coherent text that links the experiments in the main text together with the ones presented in the appendix, possibly explaining when necessary what changes in the training methods.
This would help the reader to understand your aim and follow it throughout the text and the results.

---

### Official Review · Reviewer_i356 · 2021-10-27

**Rating:** 7
**Confidence:** 4

**Review:**

This paper investigates neural networks’ preferences for synthetic features (sprites) when they are confounded during training. The study involves 5-way classification tasks where the “classes” consist of combinations of pairs of features which differ in color, shape, or texture. After training on these tasks, it is possible to evaluate what networks have learned by rendering all possible pairs of features and measuring which of the two features determines the class to which the image is assigned. The study finds no intrinsic bias toward color, shape, or texture. Instead the features that networks prefer depends on the number of pixels covered and the consistency of the feature.

This is a nice study. Its main shortcoming is that the concepts of color, shape, and texture that are investigated here are simpler than those of the 3D world, and thus it is not clear to what extent the intuition developed applies to real-world images. On the other hand, there is an inherent tradeoff between the controllability that synthetic data provides and the realism of natural images. An expanded version of this paper might explore multiple points on that Pareto frontier, but the setting investigated here seems more than adequate for a workshop paper.

Other minor comments:

- Is pixel count independent of the noise factors? If objects can be occluded, then larger objects are more likely to occlude small objects than small objects are to occlude large objects. I doubt this could explain the effect of pixel count, but it would be nice to verify that.
- I really like the idea of a model for feature preference, but I think it would be neat to have a more mathematical formulation. At least for pixel count and predictivity, it seems like it should be possible to manipulate both factors and find a curve that fits the data.

---

### Decision · Program_Chairs · 2021-11-02

Accept (Poster)